# Structure-function analysis of purified proanthocyanidins reveals a role for polymer size in suppressing inflammatory responses

Audrey Inge Schytz Andersen-Civil[1], Milla Marleena Leppä[2], Stig M. Thamsborg[1], Juha-Pekka Salminen[2] & Andrew R. Williams [1✉]

Proanthocyanidins (PAC) are dietary compounds that have been extensively studied for beneficial health effects due to their anti-inflammatory properties. However, the structure-function relationships of PAC and their mode-of-action remain obscure. Here, we isolated a wide range of diverse PAC polymer mixtures of high purity from plant material. Polymer size was a key factor in determining the ability of PAC to regulate inflammatory cytokine responses in murine macrophages. PAC polymers with a medium (9.1) mean degree of polymerization (mDP) induced substantial transcriptomic changes, whereas PAC with either low (2.6) or high (12.3) mDP were significantly less active. Short-term oral treatment of mice with PAC modulated gene pathways connected to nutrient metabolism and inflammation in ileal tissue in a polymerization-dependent manner. Mechanistically, the bioactive PAC polymers modulated autophagic flux and inhibited lipopolysaccharide-induced autophagy in macrophages. Collectively, our results highlight the importance of defined structural features in the health-promoting effects of PAC-rich foods.

[1] Department of Veterinary and Animal Sciences, University of Copenhagen, Frederiksberg, Denmark. [2] Natural Chemistry Research Group, Department of Chemistry, University of Turku, Turku, Finland. ✉email: arw@sund.ku.dk

There is currently intense interest in elucidating how dietary components may influence health and disease, in part due to increasing prevalence of chronic inflammation throughout the world[1,2]. Plant bioactive components found in the diet that may modulate inflammation include carotenoids, plant sterols, glucosinolates, and polyphenols[3]. Polyphenols represent one of the major groups of natural antioxidants with more than 8000 polyphenolic compounds identified in various plant species[4]. They are specialized plant metabolites and are commonly found in the human diet, especially in fruits and vegetables[5,6]. Polyphenols act as the active component in many commercialized plant preparations, and have been associated with beneficial effects on health[3,6]. Epidemiological and experimental studies have thus investigated how polyphenols may have protective properties against cardiovascular diseases, cancer, diabetes, neurodegenerative disease, and microbial infections, among other diseases[4,7,8].

Proanthocyanidins (PAC), also known as condensed tannins, are among the most common dietary polyphenols[9]. They are characterized by their high molecular weight, and consist of oligomers and polymers of flavan-3-ol monomeric subunits[6,10]. Flavan-3-ols are formed by two phenyl rings, A and B, and one heterocyclic ring C, with different hydroxylation patterns of rings A and B[11,12]. They may also display gallic acid residues attached by an ester bond to the C-ring hydroxyl, as seen in catechin gallates[13]. The four most common flavan-3-ols are (+)-catechin and its *cis* isomer (−)-epicatechin, and (+)-gallocatechin and its *cis* isomer epigallocatechin[10], which can be joined by inter-flavanol bonds to form oligomers or polymers. When part of an oligomeric or polymeric PAC structure, (epi)catechins are known as procyanidin (PC) units and (epi)gallocathechins are known as prodelphinidin (PD) units (Fig. 1a). Depending on the nature of their linked flavan-3-ol subunits, PAC are thus referred to as either PC-type PAC, which are the most common type of PAC, PD-type PAC, or as PC/PD mixtures[10,12]. PAC are further characterized based on their galloylation and polymerization. The percentage of galloylation describes the proportion of galloylated monomers, while the number of monomeric subunits dictates the degree of polymerization[13]. The polymer chain lengths of PAC molecules vary greatly depending on the plant sources and thus, mean degree of polymerization (mDP) is a commonly used measure for the average number of monomeric subunits.

Inflammation is a natural response to harmful stimuli, and results in the secretion of several cell-derived mediators, including cytokines, prostaglandins, and reactive oxygen species (ROS) to protect cells and tissues. The detrimental effects of inflammation arise when it perseveres and causes permanent damage of the tissues due to homeostatic imbalance of the immune regulatory functions[5]. Numerous studies have shown that PAC can have beneficial effects on inflammatory and metabolic disorders, such as obesity and inflammatory bowel disease[14,15]. Furthermore, epidemiological studies recorded lower incidences of cancer, atherosclerosis, and metabolic syndrome in human populations consuming PC-rich foods, although conclusive evidence is still lacking to support beneficial effects on human health[8,16]. The mechanisms underlying the anti-inflammatory effects of PAC are unclear, but are speculated to include direct interactions between PAC molecules and immune cell functions and pathways, as well as a potential prebiotic effect on the gut microbiota. There is a general consensus that PAC molecules remain stable in the stomach, and that only monomers are absorbed in the small intestine[17,18]. Thus, the majority of ingested PAC reach the colon, where they may be metabolized by the gut microbiota[19,20]. Moreover, polymeric PAC remain at the lumen–mucosal interface where they may also directly interact with epithelial or immune cells.

Several studies have shown that PAC can suppress several inflammatory markers in macrophages and intestinal epithelial cells in vitro, such as ROS and tumor necrosis factor-α (TNF-α), as well as the expression of inducible nitric oxide synthase and cyclooxynase-2 proteins, and thereby suppressing nitric oxide (NO) and prostaglandin $E_2$. This may be due to the downregulation of the p38 mitogen-activated protein kinase (MAPK) and Akt signaling pathways, leading to the inhibition of the transcription factor nuclear factor-κB (NF-κB), which is important for the regulation of genes involved with inflammation[21–24]. The direct effects of PAC on other cell lines, including lymphoid cell lines and peripheral blood mononuclear cells, has also been demonstrated by their ability to down-regulate transcription and secretion of pro-inflammatory cytokines and to up-regulate the secretion of anti-inflammatory cytokines[12,22,25,26]. Furthermore, anti-inflammatory properties of PAC have also been described in animal models of various diseases, where dietary PAC have been shown to modulate NF-κB sigaling and inhibit the production of ROS and pro-inflammatory cytokines such as IL-6 and TNFα[14,15,27–29].

Despite these documented effects of PAC on inflammatory activity, mechanistic and structure–function activity studies are limited. Investigation of the bioactivity of PAC is challenging due to the complexity of the PAC molecules and associated difficulties with characterization and purification. The isolation of PAC with high polymerization remains a challenge due to the difficulties related to separating polymeric molecules of different sizes from each other. Thus, structure–function relationship studies with PAC have been limited to relatively crude fractions consisting of mixtures of low or high mDP. Studies utilizing such fractions have suggested that the degree of polymerization of PAC may be an important parameter for the outcome observed in vitro and in vivo, with higher mDP often correlating with higher biological activity[22]. For example, γδ T cells are more effectively activated by PAC fractions with an mDP higher than six[30], and inhibition of lipopolysaccharide (LPS)-induced cytokine production in macrophages also appears to correlate with increasing mDP[22]. In addition, the PC/PD ratio is thought to play a role for the activity of PAC, potentially due to the ability of PD-type polymers to form greater numbers of hydrogen bonds with proteins because they have more hydroxyl groups than PCs[31].

Recently, we developed a method that splits the characteristic PAC polymeric mixtures into precise fractions by Sephadex LH-20 fractionation followed by semi-preparative liquid chromatography (LC)[32]. This resulted in chemically well-characterized PAC samples with higher purity, differing mDPs and PC/PD ratios, and thus allowed a better comprehension of the relationship between the structure and bioactivity of PAC. Here, we investigated how highly purified PAC with differing structural characteristics may affect inflammatory responses. We utilized interleukin 6 (IL-6) production from LPS-activated RAW 264.7 macrophages as a screening tool to assess structure–function relationships of PAC isolated from two different plant sources. Combined with in vitro and in vivo transcriptomics after exposure to different PAC polymers, we identified relationships between PAC structure and bioactivity that may have broad implications for the development of PAC-based dietary supplements.

## Results

**Purification of PAC and analysis of structural features.** As a first step in elucidating the structural characteristics that govern the bioactivity of PAC, we selected two plant sources to serve as source material for isolation of purified fractions. In order to assess the relative importance of PC or PD units, we selected

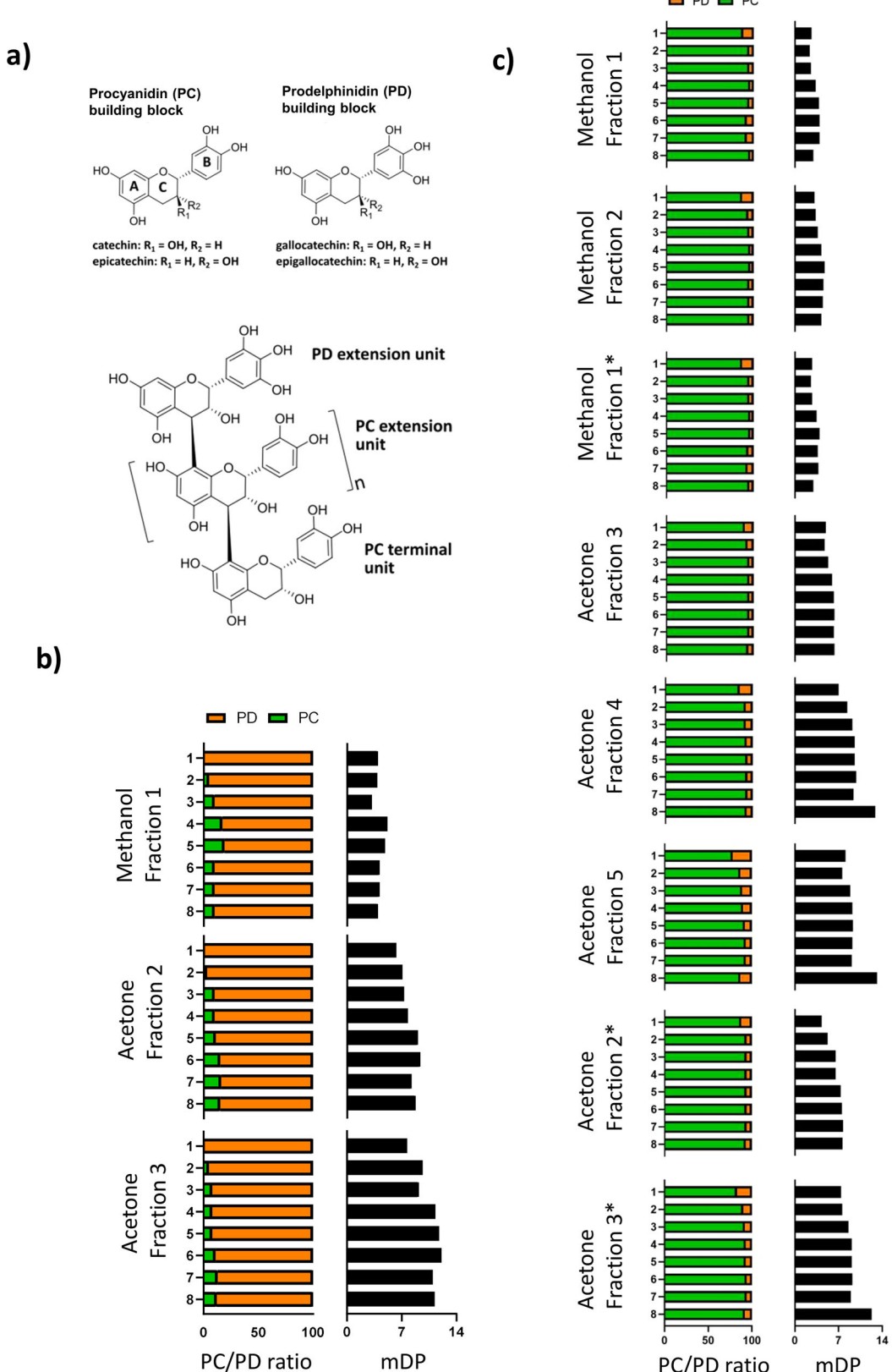

plants that contain either PC-rich PC/PD mixtures (hereafter PC-type PAC) or PD-rich PC/PD mixture (hereafter PD-type PAC). Grape (*Vitis vinifera*) pomace, hereafter GP, is a by-product of wine production that is rich in PC-type PAC. In contrast, alpine currant (*Ribes alpinum*), hereafter AC, is a temperate herb found in Northern Europe that primarily contains PD-type PAC.

Thus, material from these two distinct plant sources allowed us to derive pure PAC samples mainly consisting of either PC- or PD-type polymers, which we further sought to fractionate into samples of a precisely defined mDP. PAC were extracted with acetone/water (80/20, v/v) and fractionated by Sephadex LH-20 gel chromatography. Semi-preparative LC was subsequently used to

**Fig. 1 Molecular characteristics of proanthocyanidins. a** The structures of (epi)catechin and (epi)gallocatechin, which give basis to the most common proanthocyanidin (PAC) structural units, procyanidin (PC) and prodelphinidin (PD) units are displayed as single building blocks. The figure also depicts a model structure of an oligomeric PAC consisting of PD and PC extension units and a PC terminal unit ($n$ = number of PC or PD extension units). PAC samples used in this study showed varying chemical characteristics, with samples extracted from **b** alpine currant (*Ribes alpinum*) being rich in prodelphinidins (PD), whereas samples extracted from **c** grape (*Vitis vinifera*) pomace were rich in procyanidins (PD). Samples were purified and extracted by Sephadex LH-20 fractionation with methanol followed by acetone. Further purification was conducted by semi-preparative liquid chromatography, resulting in eight highly purified samples for each of the Sephadex fractions. Each sample was analyzed by UPLC-MS/MS samples produced from grape pomace were prepared in two batches, thus fractions labeled with "*" represent the second batch of samples. mDP mean degree of polymerization.

derive 24 and 64 highly purified samples from AC and GP, respectively[32]. Ultraperformance liquid chromatography-tandem mass spectrometry (UPLC-MS/MS) analysis of each of the isolated PAC samples was used to assess structural features[33]. The samples derived from AC were 81–99% PD-type PAC with mDP values ranging between 3.2 and 12.1. In contrast, samples derived from GP were 78–96% PC-type PAC with mDP values ranging between 2.4 and 13.2 (Fig. 1b, c). Thus, our technique successfully isolated highly purified PAC covering a wide range of mDP.

**Reduction of IL-6 secretion in LPS-activated macrophages stimulated with PAC.** To assess the relative ability of the isolated PAC to modulate inflammatory responses, murine RAW 264.7 macrophages were stimulated with LPS alone or LPS combined with purified PAC derived from GP or AC. Samples were initially screened at concentrations of 3–50 μg/mL that resulted in no toxicity to the cells (Supplementary Fig. 1), and was comparable to previous studies, which have demonstrated anti-inflammatory activity of PAC-rich extracts in cells at similar or higher concentrations[21,23]. PAC samples (15 μg/mL) derived by semi-preparative LC significantly reduced LPS-induced IL-6 secretion. For GP, samples with low mDP (< 4) inhibited IL-6 secretion inefficiently, with around 30% inhibition. Greater inhibition (up to 60%) was observed in samples with higher mDP, with a consistently high inhibition evident in a group of samples with mDPs between 5 and 10 (Fig. 2a). However, increasing mDP above 10 did not enhance activity, but resulted in a decrease of IL-6 inhibition. Thus, the relationship between mDP and inhibition of IL-6 secretion followed a quadratic relationship ($p < 0.001$ by regression analysis; Fig. 2a). This was further demonstrated by grouping samples into three distinct groups with either low, medium, or high mDPs, which showed clearly that the samples with a medium mDP had a higher ($p < 0.01$) inhibition of IL-6 secretion than those with a low or high mDP (Fig. 2b). As the different samples were tested at equal w/v amounts, the results suggest that the mDP was an important contributor to the bioactivity. Results obtained from samples with differing mDP tested at equimolar concentrations of 7.8 μM (Supplementary Fig. 2A) showed similar results, with medium mDP samples being most effective, confirming that the polymerization determined the activity. Overall, these results demonstrate that increasing the mDP of GP PAC enhanced anti-inflammatory activity up to a threshold, where after mDP no longer was associated with high activity.

Evaluation of the AC samples, comprised almost exclusively of PD-type PAC, revealed a generally lower suppression of IL-6 secretion than that achieved by GP PAC across a similar range of mDP (Fig. 2c). Again, the relationship between mDP and inhibition of IL-6 secretion was best described by a quadratic function, with the most active samples falling in the middle of the mDP range of around 6–9 (Fig. 2c), which was further confirmed by analyzing clusters of low, medium, and high mDP (Fig. 2d). This was also true for AC samples with differing mDP tested at equimolar concentrations of 7.8 μM (Supplementary Fig. 2B).

However, we noted that the magnitude of the IL-6 secretion was generally less than that achieved by GP PAC.

Thus, these data demonstrate a clear relationship between mDP and anti-inflammatory activity, with a positive correlation up until a threshold after which activity declines with increasing mDP.

**Grape PAC modulate macrophage transcriptional responses in a polymerization-dependent manner.** To investigate the mechanistic basis by which PAC regulated macrophage function, three samples derived from GP by semi-preparative LC were selected for transcriptomic experiments. Cells were stimulated with samples with mDP 2.6 (low), 9.1 (medium), and 12.3 (high), respectively, and global transcriptomic profiles were determined 6 h later. Principal component analysis demonstrated a strong influence of LPS (Fig. 3a). Moreover, a clear effect of PAC was observed that correlated closely with mDP. Consistent with the IL-6 inhibition data, the largest transcriptional changes were observed in cells exposed to medium mDP PAC, whereas PAC with high or low mDP were less effective at modulating the cellular responses both in resting cells and those activated with LPS (Fig. 3). Low mDP PAC were particularly ineffective at inducing transcriptional changes, indicating that an mDP ≥3 is necessary for optimal regulation of inflammatory activity. Moreover, cells stimulated with the high mDP PAC were also not as differentiated from their respective controls as the cells stimulated with medium mDP PAC, indicating that increasing mDP beyond a certain threshold appears to decrease bioactivity.

*Transcriptional responses in resting macrophages.* In resting cells (no LPS), high mDP PAC treatment induced the expression of genes involved with inflammatory responses, such as cytokines from the interleukin 1 family (e.g. *Il1a* and *Il1b*), as well as a number of inflammatory chemokines (*Cxcl2, Ccl22, Ccl3*) (Fig. 3b). This was supported by pathway analysis suggesting that the interferon-signaling and inflammatory pathways were upregulated (Fig. 4a). Similarly, resting cells stimulated with medium mDP PAC also showed an upregulation of chemokines (*Ccrl2, Cxcr4*) and interleukins. Consistent with this, the most upregulated pathway in resting cells stimulated with medium or high mDP PAC was the erythrocyte differentiation pathway, which includes genes encoding colony stimulating factor (*Csf2* and *Cfs3*), interleukins (*Il1a, Il3, Il6, Il9,* and *Il11*), and transforming growth factors (*Tgfb1, Tgfb2, Tgfb3*). Furthermore, both samples decreased the REACTOME cholesterol biosynthesis pathway (Fig. 4a), and upregulated the activating transcription factor 3 (*Atf3*). In contrast, treatment with low mDP PAC induced minimal transcriptional changes, with no significantly regulated gene pathways. Thus, in the absence of inflammatory stimuli, PAC with a medium-high mDP induced semi-maturation of resting macrophages, whereas PAC with low mDP lacked this stimulatory capacity.

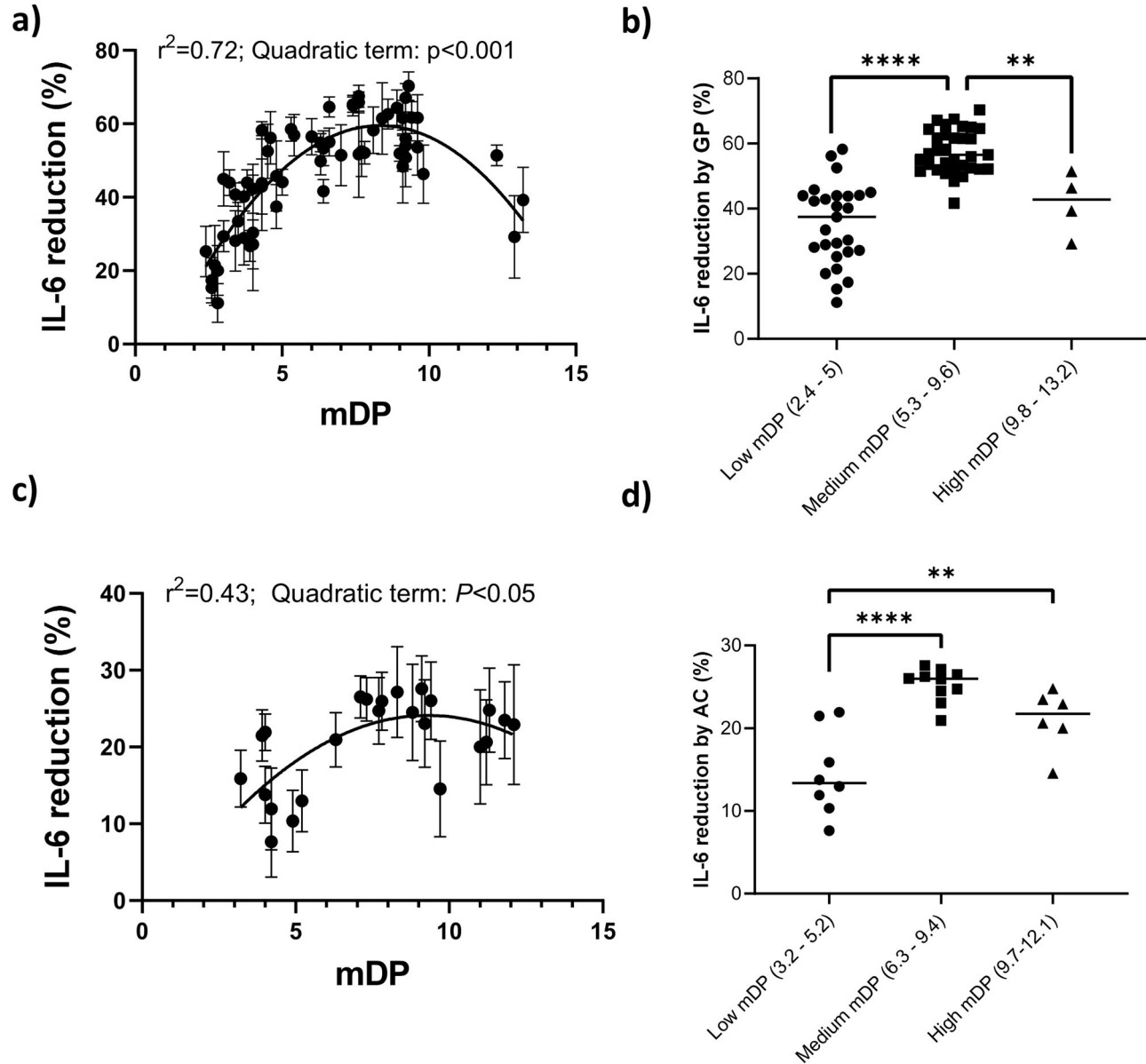

**Fig. 2 Suppression of IL-6 secretion in lipopolysaccharide-activated macrophages by proanthocyanidins is influenced by polymerization.**
**a** Proanthocyanidins (PAC) derived from grape pomace (GP) reduced IL-6 secretion in lipopolysaccharide (LPS)-activated macrophages, with the relationship between mDP and IL-6 reduction best described by a quadratic function (quadratic regression with curve-fitting analysis). Each data point represents mean and SEM of at least three independent experiments. **b** IL-6 reduction in GP PAC samples stratified into low, medium, and high mDP. **c** PAC derived from alpine currant (AC) reduced IL-6 secretion in lipopolysaccharide (LPS)-activated macrophages, with the relationship between mDP and IL-6 reduction best described by a quadratic function (quadratic regression with curve-fitting analysis). Each data point represents mean and SEM of at least three independent experiments. **d** IL-6 reduction in AC PAC samples stratified into low, medium, and high mDP. *$p < 0.05$, **$p < 0.01$, ***$p < 0.001$ by one-way ANOVA. All samples were tested at 15 μg/mL.

*Transcriptional responses in lipopolysaccharide-activated macrophages*. To explore the mechanisms underlying the anti-inflammatory activity during LPS activation, we next investigated transcriptional responses in LPS-stimulated cells. Medium mDP PAC had the most prominent impact on gene expression levels, compared to low and high mDP PAC (Fig. 3c). However, investigation of the top 15 regulated genes in LPS-activated macrophages showed that 25% of the genes were regulated by at least two of the three samples (Fig. 3d). Moreover, we found a large number of gene pathways to be regulated by medium mDP PAC in LPS-stimulated cells, whereas low and high mDP PAC regulated only two and one pathway, respectively (Fig. 4b). Despite the activation of some inflammatory pathways by PAC in resting cells, in LPS-stimulated cells we observed a strong suppression of multiple inflammatory pathways including JAK–STAT signaling, which was significantly downregulated in LPS-activated cells stimulated with medium mDP PAC (Fig. 4b).

Another characteristic of the medium mDP PAC was its ability to upregulate pathways connected to lysosome function and transferrin endocytosis. The transferring endocytosis pathway was upregulated regardless of LPS-activation (Fig. 4a, b). Genes in this pathway encode for numerous vacuolar-type H$^+$ATPases (V-ATPases) responsible for the acidification of phagolysosomes, allowing for endosome trafficking and neutralization of microorganisms[34–36]. Most of the upregulated V-ATPases belonged to the V1 Domain responsible for ATP hydrolysis,

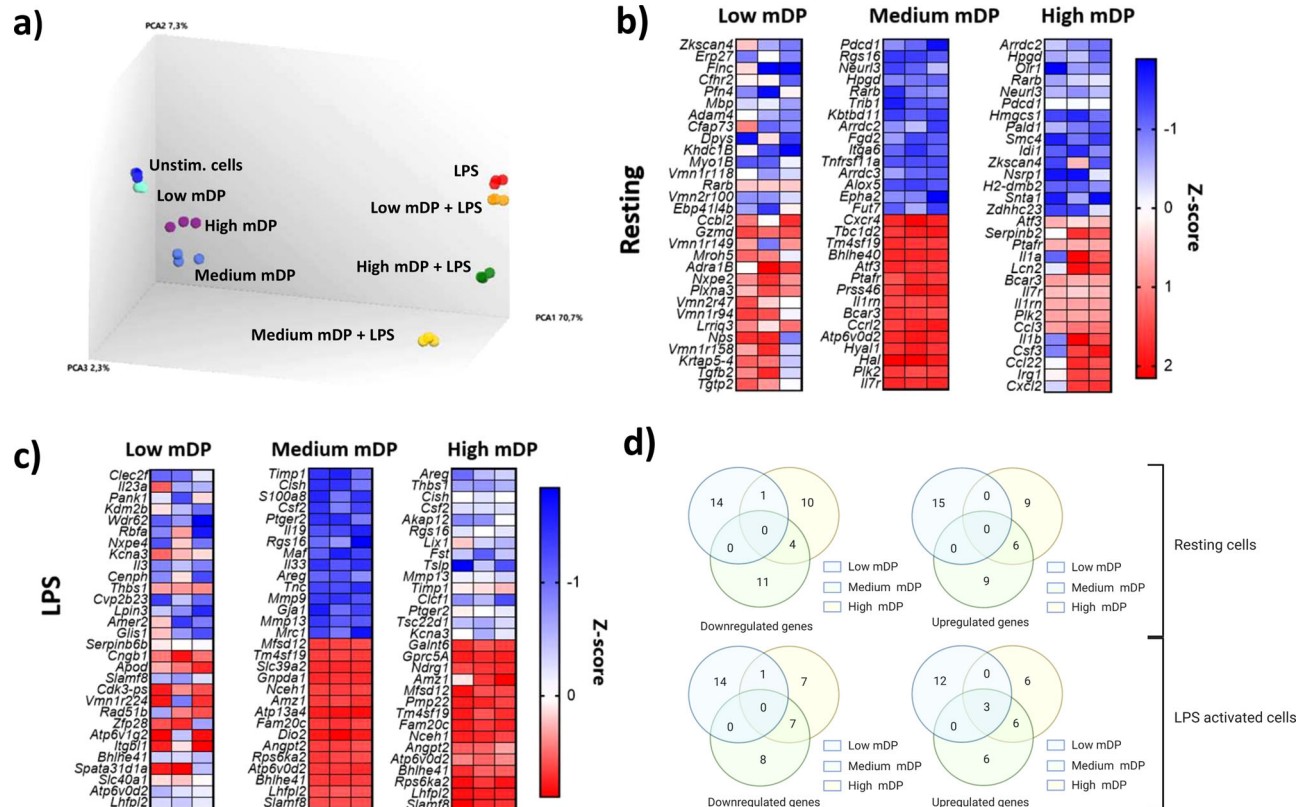

**Fig. 3 Proanthocyanidins regulate macrophage gene expression in a polymerization-dependent manner. a** Principal component analysis plot showing a clear effect of lipopolysaccharide (LPS) and proanthocyanidins (PAC). Medium mDP PAC was the most differentiated treatment from respective controls with and without LPS. The most prominent transcriptional changes were observed in cells exposed to PAC with an mDP of 9.1, whereas PAC with a higher or lower mDP were generally less effective at modulating the cellular responses. **b** Top 15 up- and down-regulated genes by PAC in resting cells (no LPS treatment). **c** Top 15 up- and down-regulated genes by PAC in LPS-activated cells. **d** The top 15 genes regulated by each PAC samples in resting or LPS-activated RAW 264.7 macrophages were investigated. Out of a total of 69 regulated genes, 25% were regulated by at least two of the PAC samples with differing mDPs in LPS-activated RAW 264.7 macrophages. Experiments were conducted with triplicate samples.

whereas fewer genes belonged to the V0 domain responsible for proton translocation. Of note, *Atp6v0d2* was the most upregulated V-ATPase gene in LPS-activated macrophages stimulated with low, medium, and high mDP PAC, and the expression level was also significantly increased in resting macrophages stimulated with medium mDP PAC (Fig. 3b, c).

Both medium and high mDP PAC significantly downregulated the cytokine inducible SH2-containing protein (*Cish*), which negatively regulates cytokine signaling of the JAK–STAT pathway[37,38], and colony-stimulating factor 2 (*Csf2*), which regulates function and differentiation of macrophages. Interestingly, all three samples significantly upregulated *Slamf8*, which has shown to negatively regulate inflammatory response, including *Nox2* activity in macrophages[39].

Collectively, these data show that the anti-inflammatory profile induced by all three samples was qualitatively similar, with a suppression of genes encoding inflammatory cytokine and signaling-related genes, and an up-regulation of genes involved in lysosome function. However, mDP had a substantial impact, with low mDP PAC being ineffective at regulating transcription, and medium mDP PAC clearly being the most efficient at regulating LPS-induced changes in gene expression.

**Grape PAC directly induce transcriptional changes in vivo in a polymerization-dependent manner.** Having established that polymerization was an important factor for direct modulation of cellular responses by PAC, we next assessed whether in vivo

administration of purified grape-derived PAC Sephadex fractions of differing mDP would also bring about direct transcriptional changes. To this end, mice were dosed repeatedly over a short time-period of ten days to assess responses in the intestine by RNA sequencing. In mice dosed with low mDP PAC, only two significantly altered genes were identified in ileal tissue compared to the control group. In contrast, a total of 27 genes were significantly modulated in tissue from mice dosed with a Sephadex fraction from which the highly active medium mDP PAC had been derived (Fig. 5a). Principal component analysis also demonstrated a clear effect of medium mDP PAC, whereas low mDP PAC were not substantially differentiated from controls (Fig. 5b). Notably, in both groups, the most upregulated gene was *S100g*, encoding the S100 calcium binding protein G, which transports $Ca^{2+}$ from the apical to the basolateral membrane of the enterocytes and sustains the essential intracellular concentration of calcium[40]. Other notable genes upregulated by medium mDP PAC included *Rspo1*, encoding R-spondin-1, a growth factor-like protein that induces crypt cell proliferation and promotes intestinal epithelial healing[41], and which has shown to be protective against intestinal damage and colitis. Downregulated genes were largely related to both inflammation and lipid metabolism, consistent with reports that acute PAC intake modulates bile acid metabolism and lipid homeostasis[42]. Notable genes included the Myd88-responsive anti-microbial peptide gene *Reg3g*, as well as the reactive free radical nitric oxide 2 (*Nos2*) and *Gata4*, a gene known to be involved in bile acid homeostasis[43]. This was

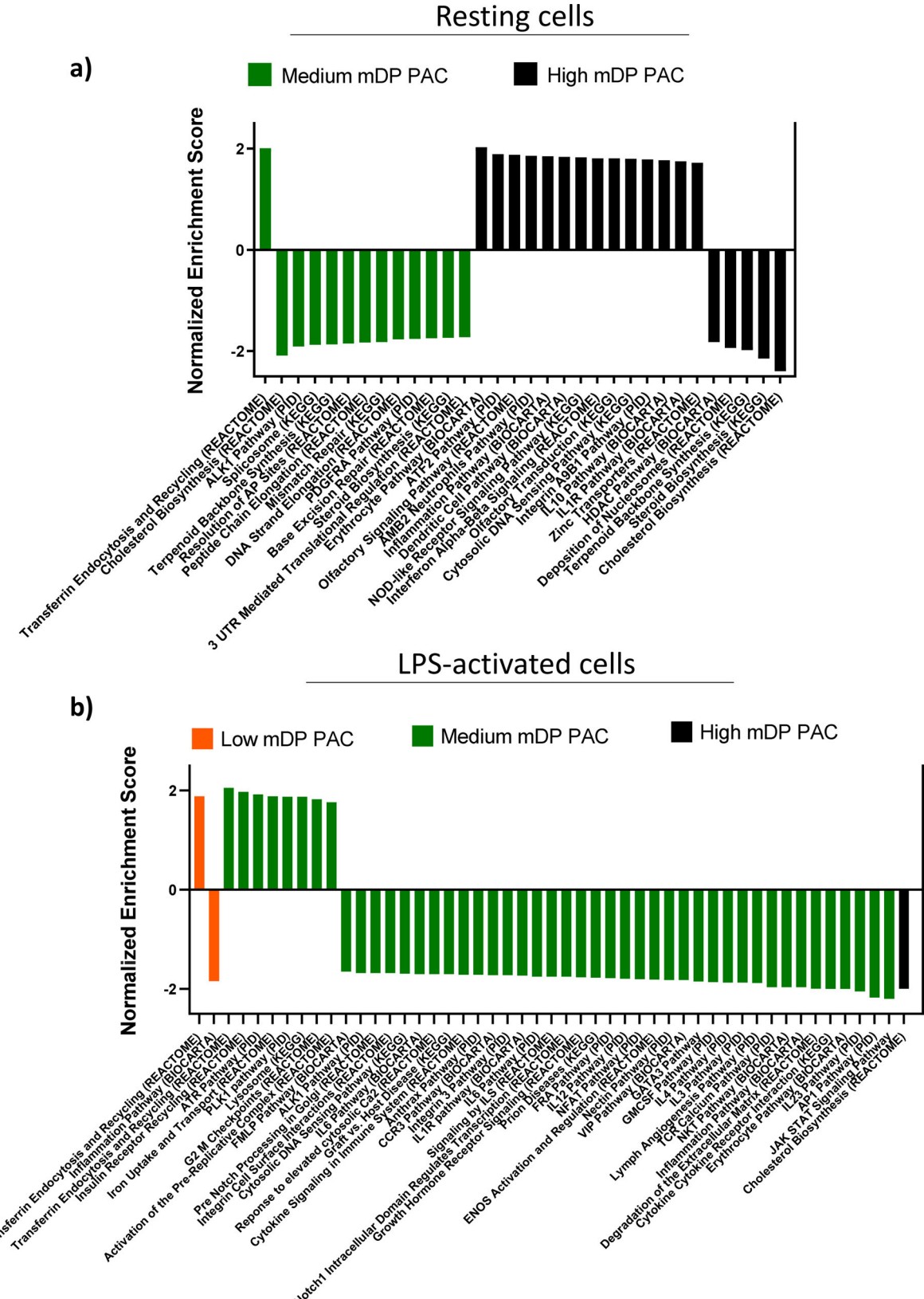

**Fig. 4 Regulation of gene pathways in RAW 264.7 macrophages stimulated with proanthocyanidins with differing degrees of polymerization. a** Gene pathways regulated by medium mDP (9.1) and high mDP (12.3) grape pomace proanthocyanidins (PAC) in resting RAW 264.7 macrophages. **b** Gene pathways regulated by low mDP (2.6), medium mDP (9.1), and high mDP (12.3) grape pomace PAC in lipopolysaccharide-activated RAW 264.7 macrophages. Experiments were conducted with triplicate samples. Shown are significantly regulated pathways (*p* value <0.001; *q* value <0.15) identified by gene-set enrichment analysis.

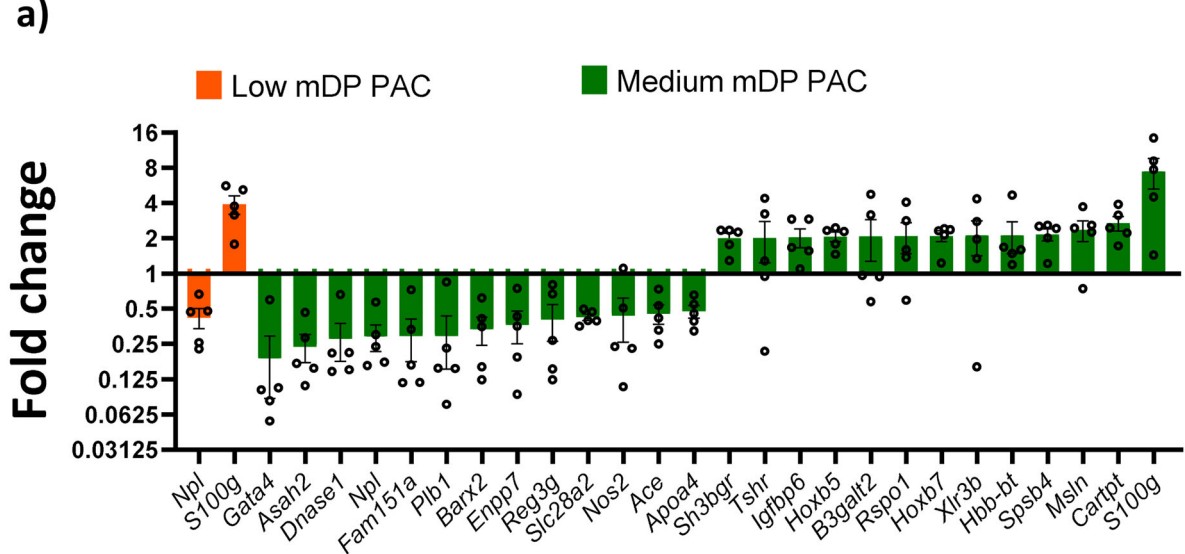

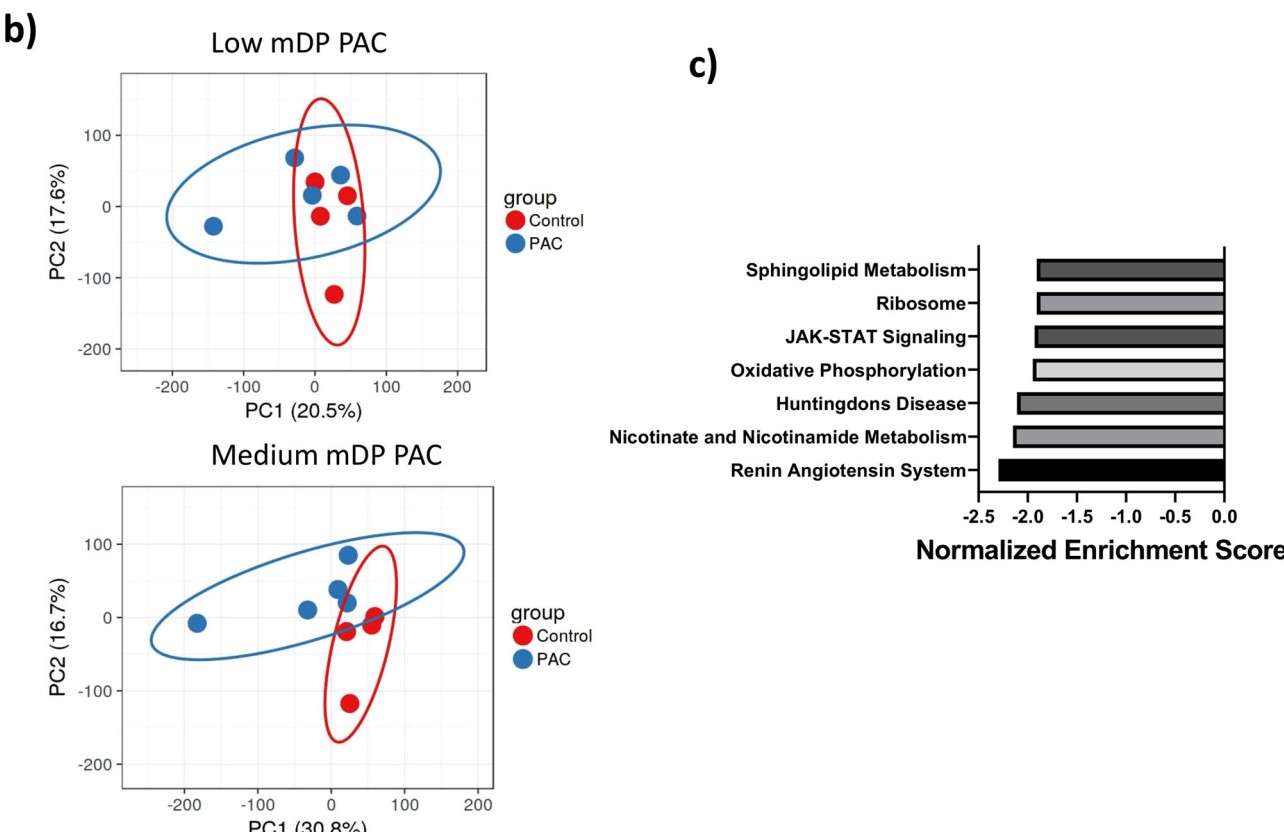

**Fig. 5 Regulation of gene expression in mouse ileum tissue by proanthocyanidins. a** Significantly regulated genes (fold change >2; *q* value >0.8 by NOIseq analysis) obtained from RNA sequencing of mouse ileum tissue in mice dosed with low mDP proanthocyanidins (PAC) or medium mDP PAC. Fold changes are relative to mice dosed with only water. *n* = 5 mice per treatment group. **b** Principal component analysis showed an effect of PAC mDP demarcated in two distinct clusters. *n* = 5 mice per treatment group. **c** Gene pathways identified in the KEGG database that were downregulated (*p* value <0.005; *q* value <0.1) by treatment with medium mDP PAC, identified by gene-set enrichment analysis.

supported by gene-set enrichment analysis which demonstrated that gene pathways including lipid metabolism and pro-inflammatory pathways (JAK–STAT signaling) were significantly suppressed (Fig. 5c). qPCR analysis confirmed the regulation of these genes in mice dosed with medium mDP

PAC, whereas responses in mice dosed with low mDP PAC were generally less differentiated from control mice (Supplementary Fig. 3). Thus, direct changes in immunological and metabolic-related transcription can be brought about in vivo by orally delivered PAC with a defined mDP profile.

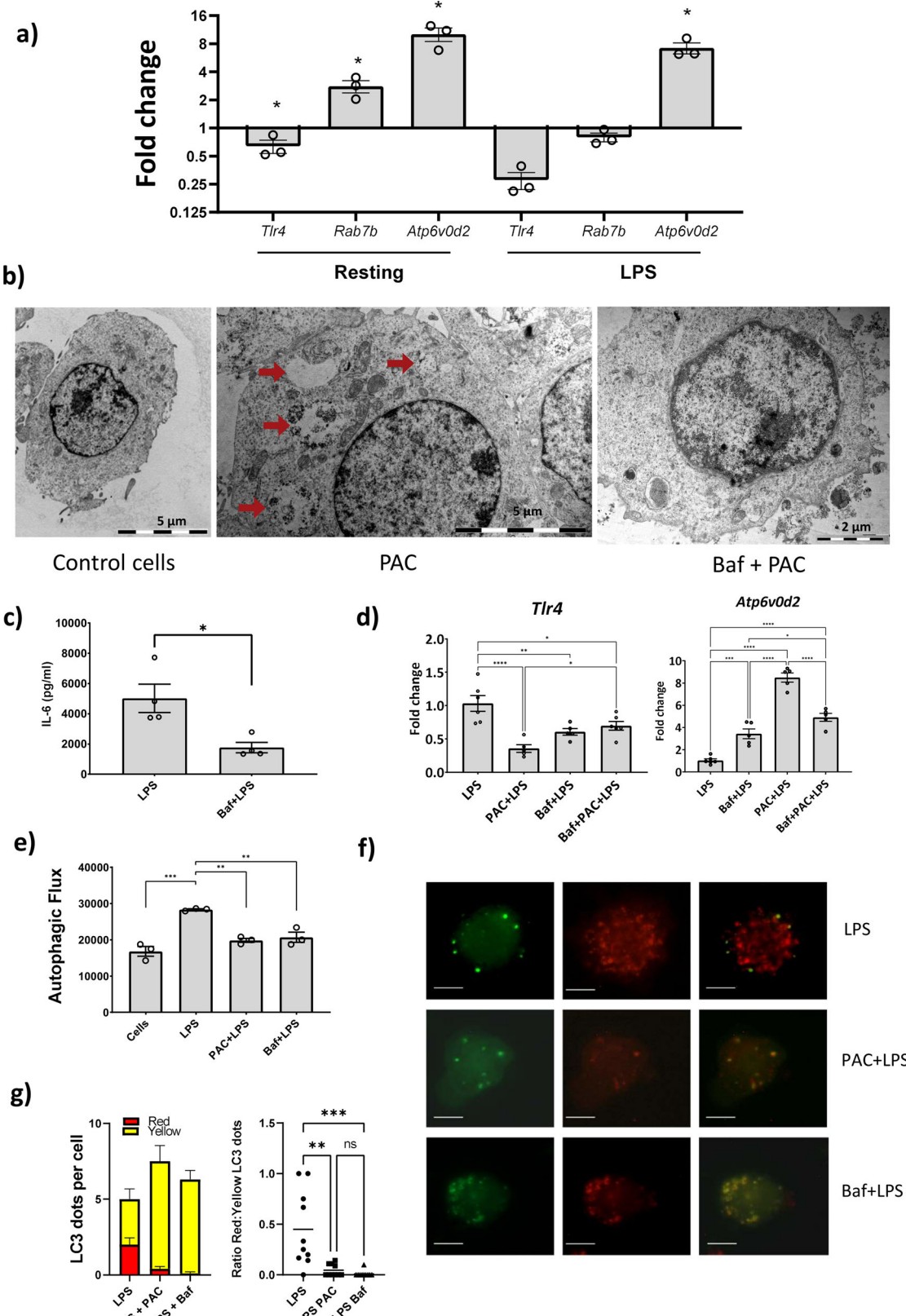

**Bafilomycin A1 antagonizes the effect of grape PAC in vitro**. We next sought to establish the mechanistic basis of the activity of the most active PAC sample with medium mDP. We noted that our in vitro transcriptomic data showed a suppression of TLR signaling pathways and an up-regulation of genes involved in V-ATPase activity and lysosome function. Therefore, we first

assessed the expression of *Tlr4*, *Rab7b*, and *Atp6v0d2* in both resting and LPS-activated macrophages by qPCR. Interestingly, we saw a significant increase in the expression of *Rab7b* in resting cells after adding GP PAC, but not in LPS-activated cells. In contrast, *Atp6v0d2* expression was upregulated in both resting and LPS-treated cells, while the expression of *Tlr4* was attenuated

**Fig. 6 Proanthocyanidins regulate gene expression, autophagosome formation, and autophagy in RAW264.7 cells. a** Expression of *Tlr4, Rab7b,* and *Atp6v0d2* in RAW264.7 macrophages exposed to proanthocyanidins (PAC) in either resting conditions or following lipopolysaccharide (LPS) stimulation. Fold changes are relative to cells exposed to medium only (resting cells) or LPS only (LPS-activated cells). Mean and SEM of three independent experiments (*$p < 0.05$ by paired *t*-test). **b** Transmission electron microscopy (TEM) images of macrophages showing formation of autophagosome-like structures in control cells or in cells incubated with PAC (15 μg/mL), or PAC combined with Bafilomycin A1 (Baf; 10 nM). **c** LPS-activated cells showed a significant decrease in IL-6 secretion when pre-stimulated with Baf (10 nM). Mean and SEM from at least three independent experiments (*$p < 0.05$ by Mann–Whitney test). **d** Expression of *Tlr4* and *Atp6v0d2* in RAW264.7 cells stimulated with LPS with PAC (15 μg/mL) and/or Baf (10 nM). Mean and SEM from at least three independent experiments (*$p < 0.05$, **$p < 0.01$, ***$p < 0.001$, ****$p < 0.0001$ by one-way ANOVA and Tukey's tests). **e** Autophagic flux in RAW 264.7 macrophages treated with LPS alone, or LPS combined with either PAC (15 μg/mL) or Baf (10 nM). Mean and SEM of three independent experiments (*$p < 0.05$, **$p < 0.01$, ***$p < 0.001$ by one-way ANOVA and Tukey's tests). **f** PAC and Baf both reduce the formation of autophagolysosomes in LPS-activated mLC3 cells. LPS-activated cells had marked abundance of red punctate spots, indicating increased autophagolysosome formation. In contrast, punctate spots and co-localization of green and red fluorescence was observed in LPS-activated cells co-stimulated with either PAC or Baf indicating an accumulation of autophagosomes and inhibition of the full autophagy pathway. Scale bar = 5 μm. **g** The abundance of red dots, i.e. autophagolysosomes, and yellow dots, i.e. autophagosomes, and the ratio between them, were enumerated in ten individual mLC3 cells (*$p < 0.05$, **$p < 0.01$, ***$p < 0.001$ by Kruskal–Wallis and Dunn's tests). Shown is mean and SEM for abundance and median for ratio.

by PAC compared to controls (Fig. 6a). However, we did not observe a decrease in either surface or intracellular expression of TLR4 protein following PAC exposure (Supplementary Fig. 4). Microscopically, we observed large numbers of vesicle-like structures in PAC-stimulated cells, which were significantly more abundant than in control cells (Supplementary Fig. 5). Electron microscopy revealed these structures to resemble autophagosomes (Fig. 6b). This may suggest either a marked increase in phagosome formation or an accumulation of autophagosomes due to altered lysosomal flux and impaired formation of autophagolysosomes (Fig. 6b). Together with the increased expression of *Atp6v0d2*, this prompted us to examine the role of V-ATPase activity in the observed effects of PAC. We co-stimulated cells with PAC and Bafilomycin A1 (Baf), a macrolide antibiotic isolated from *Streptomyces gresius*, and a potent V-ATPase inhibitor, to explore whether inhibiting V-ATPase activity would reverse the effects of PAC. Notably, we observed that co-stimulation with Baf prevented the PAC-induced accumulation of the autophagosome-like structures in the cells, suggesting that PAC may inhibit inflammatory responses by modulating V-ATPase activity (Fig. 6b and Supplementary Fig. 5).

Interestingly, we observed that Baf treatment alone inhibited IL-6 secretion (Fig. 6c) and also downregulated *Tlr4* expression in LPS-activated cells (Fig. 6d). Surprisingly, while Baf is known to inhibit the activity of the V-ATPase, expression of *Atp6v0d2* was markedly increased in Baf-treated cells, similar to that observed with PAC. Thus, Baf and PAC in isolation each induced similar changes in cellular activity, suggesting that they may act through similar mechanisms of action, and that PAC may therefore function as a V-ATPase inhibitor. However, consistent with the observed inhibitory effect of Baf on PAC-induced autophagosome accumulation, during co-incubation we observed that Baf tended to antagonize the effect of PAC. Thus, PAC-induced changes in the expression of *Tlr4* and *Atp6v0d2* were attenuated by co-stimulation with Baf (Fig. 6d). Collectively, these data show that PAC induces transcriptional and biochemical changes indicative of a modulatory role in V-ATPase, and hence phagosomal, activity.

**PAC suppress lipopolysaccharide-induced autophagy**. Given the putative association between GP PAC and V-ATPase activity, we next asked whether GP PAC would regulate autophagy, as Baf is a well-known autophagy inhibitor and other natural compounds with anti-inflammatory activity have also been shown to suppress autophagy in macrophages[44]. We found that GP PAC significantly suppressed autophagy in LPS-activated cells, to a similar extent as that seen in Baf-treated cells (Fig. 6e).

These observations were further validated using mLC3 autophagy-reporter cells. LPS-activated cells showed the characteristic appearance of autophagy with a marked abundance of red punctate spots, indicating increased autophagolysosome formation. In contrast, LPS-activated cells co-stimulated with either PAC or Baf showed punctate spots and co-localization of green and red fluorescence, indicating accumulation of autophagosomes and inhibition of the full autophagy pathway (Fig. 6f). In both PAC-treated and Baf-treated cells, the ratio of red to yellow mLC3 dots was significantly reduced compared to LPS-only treated cells (Fig. 6g). Overall, these data suggest a role for PAC in modulating inflammatory activity in macrophages by suppressing LPS-induced autophagy, in a manner analogous to the V-ATPase inhibitor Baf.

**Discussion**
The beneficial effects of PAC are numerous, and especially their ability to suppress inflammatory markers has been demonstrated in several mouse models and in vitro studies. Direct interaction with immune cells, immune cell-derived mediators, and gut microbiota may cause the prevention and alleviation of gut-related inflammation[27]. Thus, the anti-inflammatory properties of PAC have been attributed to their interaction with protein receptors and transcription factors of the NF-κB and MAPK signaling pathways, and by modulating the arachidonic acid pathway and inhibiting enzymatic activity of eicosanoid-generating enzymes[12,22,45,46]. Despite the large body of research investigating the bioactivities of PAC, their mechanism of action is still not fully understood but their anti-oxidant and anti-inflammatory activities are believed to be dependent on their varying chemical structures[5]. Thus, the bioavailability, stability, and activity of polyphenolic compounds may be correlated with the amount of hydroxyl groups, mean polymerization, and/or bond type and position between the monomers[31,47–50]. However, the complex and various molecular structures of PAC challenges the study of their structure–function in vitro. Our aim was thus to investigate how differing structural characteristics of highly purified PAC would affect bioactivity, and to explore which cellular mechanisms lead to this activity.

We initially observed a pronounced suppression of IL-6 secretion in LPS-activated murine macrophages stimulated with PAC samples derived from semi-preparative LC. Our data showed that PC-rich GP samples appeared to be more efficient at suppressing inflammatory responses in RAW 264.7 macrophages compared to PD-type PAC derived from AC. Moreover, particularly in PC-type PAC, a non-linear relationship exists between mDP and anti-inflammatory activity, with the most potent inhibition of inflammatory cytokine secretion found in polymers

with a medium (5–10) mDP. From a chemical point of view, it is worth considering the stability of compounds in, i.e., cell cultures where PD-type PAC might be prone to oxidation, which could explain the lack in bioactivity.

The stimulation of macrophages with GP PAC upregulated the expression of genes related to inflammatory responses, such as *Atf3*, which suggests that PAC are recognized by and have some activating activity in resting macrophages. This is consistent with data showing that immature human dendritic cells adopt a semi-mature state after exposure to PAC[51]. *Atf3* is induced by a variety of stimuli, such as stress, and plays an important role in immune regulation, including downregulation of TLR4, which consequently reduces the expression of several inflammatory genes, such as *Il6*, *Il12b*, and *Tnf*[52,53]. While medium and high mDP PAC tended to induce transcription of similar genes, the small number of genes that were induced by low mDP PAC were markedly different, indicating a qualitative difference in the interaction of different sized PAC polymers with macrophages.

Our data also suggest that GP PAC modulate macrophage transcriptional responses in a polymerization-dependent manner. We demonstrated that PAC with differing mDPs modulate similar genes but with varying impact on expression levels. This suggests that there is no inherent difference in the mechanism of how different size PAC affect the cell activity, but rather that the mDP may influence the ability of the cells to process the PAC molecules. Thus, we here demonstrated that GP PAC with an mDP of 9.1 had a more prominent impact on gene expression levels, especially in LPS-activated macrophages, than samples with mDPs of 2.6 or 12.3. Overall, our data demonstrated strong immunological properties of medium mDP PAC in the presence of LPS in RAW 264.7 macrophages with a downregulation of inflammatory mediators and an upregulation of anti-inflammatory mediators. The decreased bioactivity seen in low mDP PAC could be caused by limited number of structural units in PAC, thus resulting into, e.g., lower affinity towards possible target proteins as shown by Leppä et al.[50]. On the other hand, PAC with high mDP might exceed the saturation of contact points with the cells, and complicate efficient structural interactions and/or endocytosis, and thereby explain the reduced activity seen in cells stimulated with high mDP PAC.

What was also noteworthy was the prominent upregulation of JAK-STAT and inflammation pathways in LPS-activated macrophages, which were consequently down-regulated by co-stimulation with medium mDP PAC. Furthermore, *Slamf8*, which is a negative regulator of inflammatory responses, was significantly upregulated by all three samples in LPS-activated macrophages. A number of genes related to the suppression of inflammatory responses were thus regulated by PAC, including *Cish* and *Csf2*. Interestingly, our data suggest that PAC seemed to elicit an inflammatory response in resting cells by upregulating interleukins as well as chemokines; however, PAC acted as an anti-inflammatory agent in LPS-activated cells.

Sephadex LH-20 fractions were tested in vivo, and the effects of medium mDP PAC on ileum transcriptomics were substantial in a healthy mouse model compared to low mDP PAC. Here, we gavaged mice every other day for 10 days with 200 mg/kg BW of highly purified GP Sephadex fractions with either low or medium mDP PAC. A significant increase in the number of regulated genes was observed in mice dosed with medium mDP PAC compared to mice dosed with low mDP PAC. Notably, the downregulation of immune-related genes, such as *Asah2* and *Enpp7*, and the upregulation of *Rspo1*, which promotes cell proliferation of crypt cells. Taken together, our in vivo data suggest that dietary PAC, when comprised of medium mDP polymers, are able to regulate cellular functions related to nutrient metabolism and immune function in the intestinal

mucosa. As mentioned above, PAC remain relatively stable with little to no absorption or metabolism until reaching the colon[54], which supports the assumption that the observed differences in gene expression in the small intestine are a clear effect of mDP. However, it should also be considered that the metabolism of PAC by gut microbiota during in vivo studies may strongly affect bioavailability, and further challenges the association of bioactivity to specific molecular structures of PAC.

The question remains as to how and where PAC interacts with immune cells. There are some controversies concerning whether or not larger PAC molecules can be taken up by macrophages like monomers and dimers, or whether they mainly interact with the proteins and lipids on the cell membrane[55,56]. However, we have previously shown internalization of PAC with an mDP of 9.5 in dendritic cells[51]. Based on our microarray data, we hypothesized that PAC may decrease TLR4 receptors on the cell surface due to increased degradation into RAB7B promoted lysosomes[57]. However, our hypothesis was not supported by flow cytometry investigations. We observed that the KEGG Lysosome and transferrin pathway was significantly upregulated by medium mDP GP PAC; moreover, the vacuolar gene *Atp6v0d2* reached high expression levels. *Atp6v0d2* plays a key role for lysosome acidification and autophagosome–lysosome fusion and thereby ensures the appropriate turnover of intracellular organelles and infective agents[58]. The observed upregulation of V-ATPase is in coherence with previous studies, suggesting localization of PAC to lysosomes following endocytosis in human DCs[51,59]. The upregulation of the KEGG lysosomal pathway, and thereby genes encoding subunits of V-ATPase, in LPS-activated macrophages was also observed in DCs stimulated with PAC isolated from cocoa[59]. Inhibitors of V-ATPase have shown to increase NF-κB activation[60], and thus a higher expression level of V-ATPase may explain the suppression of inflammatory markers observed in our study. Based on our findings with Baf added to LPS-activated cells stimulated with PAC, our data support the notion that the intracellular activity of PAC is intimately connected to V-ATPase activity. The inhibitory effect of Baf on the expression of *Atp6v0d2* in LPS-activated cells stimulated with PAC suggests that PAC has an important role in the lysosomal pathway. However, our mechanistic experiments investigating the role of PAC during autophagy suggested a decrease in autophagy levels. This was also indicated in our studies with mLC3 cells, where we observed fewer autophagolysosomes in LPS-activated cells stimulated with PAC or Baf, compared to cells stimulated with LPS only. Taken together, these findings suggest that PAC in fact inhibits the autophagy pathway in terms of autophagolysosome formation, similarly to Baf. The enhanced expression of *Atp6v0d2* could perhaps be explained by an accumulation of lysosomes or autophagosomes in the cytoplasma, in response to the inhibition of the autophagolysosome formation. Another explanation could be that an inhibition of the ATP6V0D2 protein by PAC and Bafilomycin in isolation leads to a positive feedback-loop in the expression of the *Atp6v0d2* gene.

A large body of research suggests that a combination of phytochemicals lead to more pronounced beneficial health effects, rather than single molecules[8,25]. Synergistic effects of selected PAC samples derived by semi-preparative LC were not investigated in this study; however, we clearly demonstrated a variation in cell stimulation between PAC samples derived from the same Sephadex fraction. These findings strongly suggest an impact of PAC structure on bioactivity, and support the notion of how polymeric mixtures observed on chromatograms in fact depict numerous PAC molecules with distinctive chemical and biological properties.

In conclusion, our results highlight the importance of considering structural features of PAC-rich food sources when

examining their bioactivity in different models of inflammation. Future research should focus on the intracellular signaling pathways modulated by PAC, as well as how defined structural features may modulate the efficacy of PAC-based supplements as a tool for control of health and disease.

## Materials and methods

**Samples and materials**. The samples and materials used for this study are depicted in Supplementary Table 1.

**Ethical statement**. All experiments involving animals were conducted in agreement with the Danish legislation and the Danish Animal Experiments Inspectorate with the license number 2015-15-0201-0076.

**Purification of PAC**. Extraction of PAC from plant material was performed in accordance with the protocol from Leppä et al.[32]. We conducted a series of extraction with acetone, followed by Sephadex LH-20 separation, and semi-preparative LC. For this study, the method was applied for the isolation of PAC from AC and GP, and an overview of the purification steps is described in Supplementary Table 2.

**Extraction**. The solid phases from AC and GP were stored at 4 °C in 80% analytical acetone with shaking. The samples were filtered through a Whatman filter paper in a Büchner funnel, and the liquid phase was transferred to a container and placed for evaporation overnight (O/N). Following evaporation, the extracts were filtered again 1–3 times through a Whatman filter paper in a Büchner funnel and the water phase was stored at −20 °C. The remaining plant material was immersed in 1 L of 80/20 acetone/water (v/v) and stored at 4 °C with shaking O/N for further extractions. A total of five extractions from the plant materials were performed and each sample was analyzed by UPLC-MS/MS[33]. The chromatograms of the GP samples depicted high similarities between the samples, which were thus pooled together. Likewise, the samples derived from AC were pooled and liquid–liquid extraction with 300 mL ethyl acetate followed by liquid–liquid extraction with 300 mL butanol was performed. Finally, the solvents were removed from the samples by rotary evaporation.

**Sephadex fractionation**. The samples were mixed with Sephadex LH-20 material by magnetic stirring O/N. Following O/N mixing, the solution was poured into a Büchner funnel and all liquid material was filtered through a Whatman filter paper. A series of filtrations through the Sephadex material was performed for each sample, with five fractions of 200 mL milliQ water, five fractions of 200 mL methanol, and five fractions of 200 mL of 80/20 acetone/water (v/v). A final fraction of 400 mL acetone was also collected to wash the Sephadex material for eventual remaining compounds. In order to remove the solvents from the samples prior to freeze-drying, O/N evaporation and rotary evaporation was applied to the samples containing acetone and methanol, respectively. After the Sephadex LH-20 fractionation, a comprehensive analytical characterization was performed by UPLC-MS/MS, and the samples were subsequently freeze-dried and weighed. Eventually, the methanol and acetone fractions were selectively pooled based on the similarity of their respective chromatograms. This resulted in three Sephadex AC fractions and eight Sephadex GP fractions with weights varying between 400 and 1500 mg.

**Semi-preparative LC**. The 11 Sephadex fractions were further purified by semi-preparative LC utilizing similar fractionation procedure and instrumentation as described by Leppä et al.[32]. Approximately 100–120 mg of each Sephadex fraction was fractionated via gradient elution utilizing 0.1/99.9 formic acid/water (v/v) and acetonitrile as eluents. The gradient elution started with isocratic 5% acetonitrile following a gradient from 8 to 55% acetonitrile at 4–32 min and from 55 to 80% acetonitrile at 32–35 min and lastly finishing with column wash (80% acetonitrile) and stabilization. The flow state was set to 12 mL/min and the column used was a 150 × 21.20 mm, Gemini®10 μm,C-18, 110 Å, Axia packed, Phenomenex. The samples were collected from 5.0 to 33.0 min into 2 mL Eppendorf tubes resulting in total of 168 individual tubes. The chromatographic PAC mixture (at 280 nm) was integrated from 5 to 33 min via TargetLynx software (V4.1 SCN876 SCN 917© 2012 Waters Inc.) and the total integrated was divided into eight equal parts (Supplementary Fig. 6). The 2 mL Eppendorf tubes were then pooled accordingly resulting in total of eight semi-preparative samples per Sephadex fraction. All 88 semi-preparative samples (24 AC and 64 GP semi-preparative samples) were analyzed via UPLC-MS/MS[33] for their PC/PD composition and mDP. Semi-preparative samples were brought to water phase and freeze-dried. The weights of the samples derived by semi-preparative LC varied between 2 and 17 mg.

**UPLC-MS/MS analyses**. UPLC-MS/MS analyses were carried out with similar instrumentation and methodology as described by Engström et al.[33]. The quantitation of the PC and PD units as well as the calculation of mDP were performed

with the Engström method[33] as described by Malisch et al.[61]. Similar standards were used for the quantitation of PC and PD as well as calibration curve for the calculation of the mDP as described by Leppä et al.[32]. The stability of the mass analyzer response was monitored with frequent injections of 1 μg/mL catechin[61].

**Cell culture**. RAW 264.7 murine macrophages (ATCC TIB-71) and RAW-Difluo^TM mLC3 cells (mLC3, InVivoGen) were cultured in DMEM supplemented with 10% fetal calf serum, 100 U/mL penicillin, and 100 μg/mL streptomycin. mLC3 cells are used to report autophagosome maturation and express chimeric proteins consisting of a red fluorescence protein (RFP), a green fluorescence protein (GFP), and a membrane-bound LC3 protein[62]. The RFP is resistant to acidic environment of the autophagolysosome, while dual red and green fluorescent RFP::GFP::LC3 puncta represent the early stages of the autophagic flux. Low passage numbers (< 20) were used for all experiments. When culturing mLC3 cells, the media was supplemented with Normocin (100 μg/mL) and Zeocin (200 μg/mL, every second passage). For experimental purposes, neither Normocin nor Zeocin was added to the media.

**Cytotoxicity testing**. Cytotoxicity of the isolated PAC used for the stimulation of murine macrophages was assessed using neutral red assays (Sigma-Aldrich) according to the manufacturer's instructions.

**In vitro screening of anti-inflammatory activity of PAC**. Macrophages were plated out on 96-well plates at a concentration of $2.5 \times 10^5$ cells/mL. Cells were allowed to adhere for 2 h before LPS stimulation (500 ng/mL). Where appropriate, cells were pre-incubated with AC or GP PAC or vehicle control (PBS) 30 min prior to LPS addition. After 24 h of incubation, the supernatant was collected and frozen at −20 °C, and IL-6 secretion was assessed by enzyme-linked immunosorbent assay (R&D Systems). For equimolar tests, the molarity of samples was calculated based on PC/PD ratios and mDP.

**Flow cytometry**. Macrophages were recovered using Accutase (Sigma-Aldrich), washed, and then surface staining or intracellular staining (following fixation with 4% paraformaldehyde and treatment with 0.1% saponin) was carried out at 4 °C for 30 min. TLR4 expression was detected using rat anti-mouse TLR4-APC (Clone SA15-21; Cat# 145405, BioLegend). In total, 10,000 cells were acquired on an Accuri C6 flow cytometer and analyzed using Cflow software (BD Biosciences).

**Stimulation with Bafilomycin A1**. As described above, murine macrophages were plated out on 24-well plates at a concentration of $2.5 \times 10^5$ cells/mL. Cells were allowed to adhere for 2 h before they were treated in the following order with Baf (10 nM), PAC (15 μg/mL), and LPS (500 ng/mL) with 30 min incubation time between each treatment. Stimulated cells and appropriate controls were left for incubation for 6 h before RNA extraction.

**Assessment of autophagy**. Autophagy in RAW 264.7 macrophages was assessed by measuring autophagic vacuoles with the Autophagy Assay Kit (Abcam ab139484) according to the manufacturer's protocol. In brief, cells were plated out in a 96-well plate and stimulated with either Baf (10 nM) or GP PAC (15 μg/mL) followed LPS stimulation (500 ng/mL) where appropriate. After 20 h of incubation, the cells were trypsinized, washed by centrifugation, and the pellet was re-suspended in indicator-free cell culture medium containing 5% FBS. The cells were then incubated with staining solution for 30 min at RT in the dark, followed by a series of washing steps, before analysis by flow cytometry.

**Preparation of mLC3 cells for fluorescence microscopy**. mLC3 cells were plated out on six-well plates containing coverslips, at a concentration of $2.5 \times 10^5$ cells/mL. Cells were allowed to adhere for 2 h before LPS stimulation (500 ng/mL). Where appropriate, cells were pre-incubated with either Baf (10 nM) or GP PAC (15 μg/mL) 30 min prior to LPS addition. After 18 h of incubation, the cells were washed twice with PBS and fixed for 15 min with 4% formaldehyde, followed by another washing step with PBS. Finally, the coverslips were recovered and placed onto a slide for microscopy using a Leica DM 5000B Fluorescence Microscope. Images were prepared using ImageJ software (National Institutes of Health, Bethesda, MD, USA). Image processing consisted only of adjustment of brightness and contrast. In order to ensure adjustments were equally applied to all samples, images were processed in stacks.

**Transmission electron microscopy**. Murine macrophages were plated out on 24-well plates at a concentration of $5 \times 10^5$ cells/mL. Cells were allowed to adhere for 2 h before they were treated as appropriate in the following order with Baf (10 nM), and/or GP PAC (15 μg/mL), and/or LPS (500 ng/mL) with 30 min incubation time between each treatment. The cells were incubated for 24 h before fixation and preparation for transmission electron microscopy. Cells were fixed with 3% glutaraldehyde (Merck, 1042390250) in 0.1 M mNa-phosphate buffer (pH 7.4), post-fixed in 1% osmium tetroxide in 0.1 M Na phosphate buffer, dehydrated

stepwise in a graded ethanol series, intermediate solution propylene oxide followed by pure epon O/N and next day, embedded in Epon (TAAB, T031). Semi-thin (2 μm) sections were cut with glass knifes (KnifeMaster II, LKB Bromma 7800) on an ultramicrotome (Leica Ultracut, Leica Microsystems, Wetzlar, Germany), stained firstly with 1% toluidine blue (VWR 34187.185) in 0.1% Borex (VWR 27727.231). Ultra-thin (50–60 nm) sections were sectioned with a diamond knife (Jumdi, 2 mm) on an ultramicrotome (Leica Ultracut), contrasted with 2% uranyl acetate (Polyscience, 21447) and lead citrate(Reynold, 1963), and examined using a Philips CM100 transmission electron microscope operating at 60 kV. Photographs were taken using Olympus Morada 11 megapixel camera and iTEM software (Olympus).

**RNA extraction from macrophages and transcriptomic analysis**. LPS-activated and non-activated RAW 264.7 macrophages were cultured with appropriate treatments as described above, and harvested for RNA extraction after 6 h of incubation using RNAeasy kits (Qiagen). cDNA was synthesized from 500 ng of RNA using Quantitect Reverse Transcriptase kits (Qiagen). Transcriptomic analysis was performed using the GeneChip WT PLUS Reagent kit (Thermo Fisher Scientific, CA, USA) and Affymetrix mouse Clariom S HT 24-array plate pipeline (Eurofins AROS, Denmark), with array plate processing carried out on a GeneTitan Instrument (Thermo Fisher Scientific). Transcriptome Analysis Console (TAC) software (Thermo Fisher Scientific) was used to analyze microarray data, and additional pathway analysis performed using Gene Set Enrichment Analysis software (GSEA; Broad Institute, MA, USA), using values of FDR $q$ value <0.2 and nominal $p$ value <0.02 to assess significance. qPCR was performed using perfeCTa SYBR green fastmix (Quanta Bioscience) using the following program: 95 °C for 2 min followed by 40 cycles of 15 s at 95 °C and 20 s at 60 °C. Primer sequences are listed in Supplementary Table 3.

**Treatment of mice with PAC fractions**. Six-week-old female C57BL/6JOlaHsd mice (Envigo, the Netherlands) were distributed into three groups of five mice. The mice were allowed 1 week of acclimatization, and were fed with a purified control diet (13 kJ% fat; ssniff Spezialdiäten GmbH, Germany) throughout the entire study. Each group was orally gavaged every second day for 10 days with either low (3.5) mDP GP PAC, medium (8.3) mDP GP PAC (200 mg/kg BW) Sephadex LH-20 fractions dissolved in distilled water, or water alone. Mice were euthanized by cervical dislocation, and tissue collected from the ileum was preserved in RNAlater for RNA-sequencing and qPCR. Primer sequences are listed in Supplementary Table 3.

**RNA sequencing of mouse intestinal tissue**. RNA was extracted from ileal tissue as described above. Libraries were prepared and sequenced (paired-end reads of 100 bp) on a BGISEQ-500 sequencer (BGI, Copenhagen, Denmark). Quality control was performed using SOAPnuke v1.5.2 (github.com/BGI-flexlab/SOAPnuke) to remove adaptors, reads with unknown bases more than 10%, and low-quality reads. Clean reads were mapped to the mouse genome (mm10) using Bowtie2 (v2.2.5). Differentially expressed genes were detected using NOIseq, using a Q-value for significance of >0.8 and a fold change of at least two[63]. Pathway analysis was conducted using GSEA. Predicted genes and those with no gene symbol assigned were excluded from analysis. Principal component analysis was performed using ClustVis[64]. For qPCR validation, cDNA synthesis and PCR was performed as above.

**Statistical analysis**. Data analysis was performed using the statistical softwares GraphPad Prism 7 (GraphPad Inc.) and SPSS (IBM SPSS Statistics 27). D'Agostino–Pearson and Shapiro–Wilk normality tests were used to assess data sets for normal distribution. Parametric data were analyzed using $t$-tests or one-way ANOVA analysis and Tukey's multiple comparisons test. If the data did not follow a Gaussian distribution, non-parametric tests (Mann–Whitney tests or Kruskal-Wallis tests followed by Dunns post hoc test) were applied. Regression analysis was conducted using the curve-fit option in SPSS, applying both linear and quadratic analyses.

**Reporting summary**. Further information on research design is available in the Nature Research Reporting Summary linked to this article.

## Data availability

Microarray data are available at GEO, under the accession number GSE167063, and raw GSEA data are available in Supplementary Data 1. RNA-seq data are available at GEO under accession number GSE168138, and raw GSEA data are available in Supplementary Data 2. Source data and images underlying the presented graphs are available in Supplementary Data 3 and 4.

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

## Acknowledgements
The authors would like to thank Mette Marie Arnt Schjelde for excellent technical assistance during laboratory work and Anne Koivuniemi for the maintenance of the UPLC instrument. Figure 3d was created with BioRender.com. This work was funded by the Independent Research Fund Denmark (Grant # 7026-0094B).

## Author contributions
A.I.S.A.-C. conceived research, designed and performed experiments, carried out data analysis, and wrote the manuscript. M.M.L. and J.-P.S. planned, supervised, and assisted with the chemical analyses. S.M.T. helped supervise the research and interpret the data. A.R.W. conceived and supervised the research, carried out data analysis, and wrote the manuscript. All authors participated in discussion of the data and editing the final manuscript.

## Competing interests
The authors declare no competing interests.
