## [Peer Review File · Communications Biology]

Reviewers' comments:

Reviewer #1 (Remarks to the Author):

The paper presented is a strong, well-organized study elucidating the influence of PAC structure on anti-inflammatory function in an in vitro and in vivo model. These findings contribute to the field of bioactives research in a meaningful way, though a few small issues must be addressed in order to strengthen the paper:

- Toxicity data and flow cytometry data must be included Supplemental Material
- In figure 2, the statistical analyses should be more clear. While explained in the "Statistical Analysis" section, this does not translate to the figures. Please be explicit about tests used and indicate any differences on the graphs. As it stands, it is not possible to confirm claims about whether given fractions were more effective than others, as the error bars are quite large.
- Some issues should be addressed in the discussion, particularly with respect to bioavailability and metabolism of these compounds and how this can affect differences in results between in vitro vs. in vivo models. Please expand upon this, and the implications it has on the application of these treatments.

Reviewer #2 (Remarks to the Author):

PACs were extracted from alpine currant (*Ribes alpinum*), which is rich in prodelphinidins (PD), or grape (*Vitis vinifera*) pomace, which is rich in procyanidins (PD). Sephadex L-20 fractionation followed by semi-prep LC yielded 8 highly purified samples that were characterized by UPLC-MS/MS prior to in vitro and in vivo assays. The study aimed to investigate how PACs with differing mean degrees of polymerization (mDP) contributed to bioactivity in vitro and in vivo and the cellular mechanisms involved.

Minor edits/comments

Lines 94 – 96: most PAC molecules reach the colon, references should be included.

Line 99: PACs have been shown to suppress inflammatory cytokines and ROS in vivo and these references warrant inclusion.

Major concerns

The data is interesting but appropriate quantifications and statistical analyses are absent for several figures making it difficult to evaluate the data or have confidence in the conclusions.

Figure 2: There are no statistical analyses for any of the data presented, therefore it is unclear how the authors differentiated between treatments. Without statistics it cannot be said that "the samples with the highest mDP had lower activity than those with an mDP in the 6-10 range" (lines 168 - 169). Figures do not indicate if error bars represent SD or SE. Based on the visual overlapping of error bars, data in Figure 2 do not support a relationship between mDP and IL-6 reduction.

The PCA plot shown in Figure 3A does suggest differences among PACs with low, medium, and high mDP. For Figures 3A and 5B, instead of using PCA, linear discriminant analysis (LDA) might be better to maximize separability among groups.

Figure 6B – The authors present images of structures described as "resembling phagolysosomes". It is not clear how many cells were imaged to come to the stated conclusions. Authors should provide quantification of autophagolysosome-like structures present in control, PAC-treated, and Baf + PAC treated cells.

Figure 6E quantifies autophagic vacuoles in RAW 264.7 macrophages analyzed by flow cytometry. Then the results refer to mLC3 reporter cells being used for fluorescence microscopy however,

details of what the red and green fluorescence corresponds to or how these cells are used to monitor autophagic flux are not adequately explained. Figure 6F (lines 310 – 312) further does not report how many mLC3 autophagy-reporter cells were visualized and how many punctate spots were observed/ quantified. Without quantification it is difficult to interpret the data.

Response to Reviewers

Reviewer #1 (Remarks to the Author):

1) Toxicity data and flow cytometry data must be included Supplemental Material.

Toxicity data and analysis of TLR4 expression by flow cytometry was added as supplementary material (Supplemental Figures 1 and 4 in the revised version).

2) In figure 2, the statistical analyses should be more clear. While explained in the "Statistical Analysis" section, this does not translate to the figures. Please be explicit about tests used and indicate any differences on the graphs. As it stands, it is not possible to confirm claims about whether given fractions were more effective than others, as the error bars are quite large.

We thank the reviewer for their suggestion and fully agree that more rigorous statistical analysis is required. We have taken two distinct approaches to address this issue. First, we have performed regression analysis to demonstrate a statistically significant quadratic relationship between mDP and IL-6 inhibition for both types of PAC, indicating that the correlation is non-linear and tends to decrease at high levels of mDP. Furthermore, we performed a stratification of samples based on mDP, and compared these by ANOVA. This analysis also showed that the samples with a 'medium' mDP were significantly more effective at reducing IL-6 secretion than 'low' or 'high' mDP samples. The statistical tests used for each figure has been specified in the figure legends for all figures in the manuscript. The updated graphs are presented in the revised Figure 2, as shown below:

3) Some issues should be addressed in the discussion, particularly with respect to bioavailability and

metabolism of these compounds and how this can affect differences in results between in vitro vs. in vivo models. Please expand upon this, and the implications it has on the application of these treatments.

We have now added the following additional text in the discussion on lines 401-406 in the revised version:

“ PAC remain relatively stable with little to no absorption or metabolism until reaching the colon ⁵⁴, which supports the assumption that the observed differences in gene expression in the small intestine are a clear effect of mDP. However, it should also be considered that the metabolism of PAC by gut microbiota during *in vivo* studies may strongly affect bioavailability, and further challenges the association of bioactivity to specific molecular structures of PAC.”

Reviewer #2 (Remarks to the Author):

1) Minor edits/comments

Lines 94 – 96: most PAC molecules reach the colon, references should be included

The following text was added at lines 95-96 of the revised version:

“Thus, the majority of ingested PAC reach the colon, where they may be metabolized by the gut microbiota^{19,20}”

2) Line 99: PACs have been shown to suppress inflammatory cytokines and ROS in vivo and these references warrant inclusion.

The following text was added at lines 107-109 of the revised version:

“Furthermore, anti-inflammatory properties of PAC have also been described in animal models of various diseases, where dietary PAC have been shown to modulate NF- κ B signaling and inhibit the production of ROS and pro-inflammatory cytokines such as IL-6 and TNF α ^{14,15,27–29}.”

3) Major concerns

The data is interesting but appropriate quantifications and statistical analyses are absent for several figures making it difficult to evaluate the data or have confidence in the conclusions.

We thank the reviewer for pointing this out and agree that some more data analysis and quantification is required. We have now included the requested analyses.

Figure 2: There are no statistical analyses for any of the data presented, therefore it is unclear how the authors differentiated between treatments. Without statistics it cannot be said that “the samples with the highest mDP had lower activity than those with an mDP in the 6-10 range” (lines 168 - 169). Figures do not indicate if error bars represent SD or SE. Based on the visual overlapping of error bars, data in Figure 2 do not support a relationship between mDP and IL-6 reduction.

We agree with the reviewer that the previous presentation of the data was inadequate, and we have now used both regression and ANOVA analyses to define the relationship between mDP and IL-6 reduction. We would refer the reviewer to the above response to comment 2) from reviewer #1, for a more detailed description of these changes.

The PCA plot shown in Figure 3A does suggest differences among PACs with low, medium, and high mDP. For Figures 3A and 5B, instead of using PCA, linear discriminant analysis (LDA) might be better to maximize separability among groups.

We thank the reviewer for their suggestion. As the reviewer points out, the current PCA approach indicates a clear difference between the samples, and as such have elected to stick with this analysis.

Figure 6B – The authors present images of structures described as “resembling phagolysosomes”. It is not clear how many cells were imaged to come to the stated conclusions. Authors should provide quantification of autophagolysosome-like structures present in control, PAC-treated, and Baf + PAC treated cells.

We thank the reviewer for this suggestion. To address this issue, we have now made a quantitation of the structures that were observed in the cells by standard microscopy (Supplementary Figure 5 in the revised version, and shown below), which shows a clear effect of PAC, which is abrogated by Baf treatment. The TEM pictures in the Figure 6B thus serve to provide a high magnification image of the structures, in order to confirm their appearance as autophagosomes.

A)

B)

Control

PAC

Baf+PAC

Figure 6E quantifies autophagic vacuoles in RAW 264.7 macrophages analyzed by flow cytometry. Then the results refer to mLC3 reporter cells being used for fluorescence microscopy however, details of what the red and green fluorescence corresponds to or how these cells are used to monitor autophagic flux are not adequately explained. Figure 6F (lines 310 – 312) further does not report how many mLC3 autophagy-reporter cells were visualized and how many punctate spots were observed/ quantified. Without quantification it is difficult to interpret the data.

To address this issue, we have now quantified the ratio of red to yellow dots in the three treatment groups (ten cells were counted for each treatment group) These results and statistical analysis are now presented in Figure 6 (Figure 6G in the revised version, and shown below).

g)

Furthermore, clarification on the representation of red and green fluorescence was added in the figure legend with the following text:

“LPS-activated cells had marked abundance of red punctate spots, indicating increased autophagolysosome formation. In contrast, punctate spots and co-localization of green and red fluorescence was observed in LPS-activated cells co-stimulated with either PAC or Baf indicating an accumulation of autophagosomes and inhibition of the full autophagy pathway”

In addition, further information on the mLC3 cell line and how it is used to detect autophagic flux was also added in the Material and Methods section, on lines 515-520 in the revised version.

“mLC3 cells are used to report autophagosome maturation and express chimeric proteins consisting of a red fluorescence protein (RFP), a green fluorescence protein (GFP), and a membrane-bound LC3 protein⁶². The RFP is resistant to acidic environment of the autophagolysosome, while dual red and green fluorescent RFP::GFP::LC3 puncta represent the early stages of the autophagic flux.”

REVIEWERS' COMMENTS:

Reviewer #1 (Remarks to the Author):

The authors have adequately addressed all of my concerns. This paper is appropriate for publication in Communications Biology.

Reviewer #2 (Remarks to the Author):

My comments have been addressed and I recommend the publication of this manuscript.